

# Data-driven models for flood prediction in an ungauged karst wetland: Napahai wetland, Yunnan, China

Xiao Li[1] and Jie Li[2,3]

[1] School of Statistics and Mathematics, Yunnan University of Finance and Economics, Kunming, Yunnan, China
[2] Institute of International Rivers and Eco-security, Yunnan University, Kunming, Yunnan, China
[3] Yunnna Key Laboratory of International Rivers and Transboundary Eco-security, Yunnan University, Kunming, Yunnan, China

## ABSTRACT

Flood prediction for ungauged karst wetland is facing a great challenge. How to build a wetland hydrological model when there is a lack of basic hydrological data is the key to dealing with the above challenge. Napahai wetland is a typical ungauged karst wetland. In ungauged wetland/condition, this article used the wetland open water area (OWA) extracted from Landsat remote sensing images during 1987–2018 to characterize the hydrological characteristics of Napahai wetland. The local daily precipitation in the 1987–2018 rainy season (June–October) was used to set the variables. Based on the following hypothesis: in the rainy season, the OWA of the Napahai wetland rises when there is an increase in accumulated precipitation (AP), two data-driven models were established. The study took the area difference (AD) between two adjacent OWAs as the dependent variable, the accumulated precipitation (AP) within the acquisition time of two adjacent OWAs, and the corresponding time interval (TI) of the OWA as explanatory variables. Two data-driven models (a piecewise linear regression model and a decision tree model) were established to carry out flood forecasting simulations. The decision tree provided higher goodness of fit while the piecewise linear regression could offer a better interpretability between the variables which offset the decision tree. The results showed that: (1) the goodness of fit of the decision tree is higher than that of the piecewise linear regression model (2) the piecewise linear model has a better interpretation. When AP increased by 1 mm, the average AD increased by 2.41 ha; when TI exceeded 182 d and increased by 1 d, the average AD decreased to 3.66 ha. This article proposed an easy decision plan to help the local Napahai water managers forecast floods based on the results from the two models above. In addition, the modelling method proposed in this article, based on the idea of difference for non-equidistant time series, can be applied to karst wetland hydrological simulation problems with data acquisition difficulty.

Corresponding author
Jie Li, jli_1984@hotmail.com

## INTRODUCTION

Floods are one of the most devastating disasters, causing about 620 billion US$ of economic and property losses each year (*Koks, 2018*). Nearly half of the world's population lives around rivers, lakes, and wetlands, which are at a high risk of being affected by floods (*Jongman, 2018*). Karst wetlands have been widely developed around the world (*Sullivan et al., 2014*), and floods occur frequently (*Hill & Polyak, 2020*). Due to the complex hydrological interaction mechanism formed by groundwater and surface water on the dissolution and precipitation of soluble rocks, erosion and deposition, as well as gravity collapse, accumulation, *etc.*, its flood disaster is unpredictable (*Bates et al., 2021*). In the current situation of frequent extreme climates, it is easier to expand the losses and impacts caused by such disasters, which is one of the main problems faced by the current natural disaster forecast and management (*Noone et al., 2017*; *Blöschl et al., 2019*).

The Yunnan-Guizhou Plateau in southwest China has developed a large number of karst landform units, and many areas have no features of surface water at all (*Guo et al., 2013*). If cracks or pipes in the limestone do not drain recharge quickly enough during heavy or prolonged rainfall, it can cause groundwater to overflow from the pipe network above ground, creating wetlands in low-lying areas (*Drew, 2008*; *Hill & Polyak, 2020*). In addition, under the influence of the monsoon climate, the precipitation in the rainy season is relatively concentrated, and it is easy to form a large area and periodic wetland flooding (*Long et al., 2014*). The Napahai Karst Wetland, also located in the northwestern part of Yunnan, China, has suffered repeated floods over the past 20 years, and many roads and houses in natural villages have been submerged, causing great economic losses. The reason for this is the rapid accumulation of water in low-lying areas of the catchment caused by continuous high-intensity rainfall (*Li et al., 2013*). How to predict the flooding of Napahai karst wetlands efficiently and accurately is a key issue in current flood management.

Current modeling methods for karst wetland hydrology can be divided into three broad categories (*Hartmann et al., 2014*; *Basu, Morrissey & Gill, 2022*): physics-based distributed hydrodynamic models, data-driven models, and the semi-distributed models between the two categories above (*Ghasemizadeh et al., 2012*; *Kovács & Sauter, 2007*). Among them, distributed and semi-distributed hydrodynamic models can simulate detailed hydrological processes by defining hydraulic parameters and system states, and they perform very well in simulating groundwater-surface water interactions in karst basins (*Abusaada & Sauter, 2013*; *Gill et al., 2020*; *Hartmann et al., 2014*). However, they have higher requirements for multi-level data and information (*Chen & Goldscheider, 2014*; *Gill et al., 2013*). For karst areas, it is difficult to simulate and restore the unique hydrological structures such as underground rivers and karst caves, which is the basis for applying distributed and semi-distributed hydrodynamic models based on physical processes. Thus, the value of such highly parameterized models is a recurrent debate in hydro(geo)logy (*Beven, 2006*). For the Napahai karst wetlands lacking basic hydrological monitoring such as water level monitoring, the above models are not ideal for flood forecasting.

In recent years, data-driven models for hydrological prediction of karst wetlands have mainly focused on machine learning methods. For example, *Al-Fugara et al. (2020)* and
*Naghibi, Ahmadi & Daneshi (2017)* compared the predictive power of several different machine learning methods for spatial mapping of groundwater potential. *Kurtulus & Razack (2007)* and *Lee & Tuan Resdi (2016)* used meteorological data such as precipitation to simulate water output through neural network modeling. *Guzman, Paz & Tagert (2017)* used historical data on precipitation and groundwater levels to predict future groundwater levels; *Wunsch, Liesch & Broda (2018)* used precipitation and air temperature to predict groundwater levels. The modeling methods of the latter two are also neural networks. *Basu, Morrissey & Gill (2022)* used both nonlinear time series and support vector machine methods for karst flood forecasting. The main advantage of machine learning methods is that they usually have a good prediction accuracy without understanding in depth the hydrological mechanism (*ASCE, 2000a*; *ASCE, 2000b*); the disadvantage is that the output results do not provide any model expression, which is an added challenge for outcome interpretability (*Lee & Tuan Resdi, 2016*).

Moreover, in order to obtain high goodness of fit, data-driven models usually need abundant data for model construction and validation. However, ungauged karst wetlands, widespread all over the world, lack in basic hydrological monitoring, and are located in places where it is challenging to collect sufficient date to construct data-driven models (*Malagò et al., 2016*). Due to these challenges, it makes sense to construct models for flood prediction in these areas.

Previous hydrological studies in Napahai wetland have verified that there is a strong correlation between accumulated precipitation (AP) and the open water area (OWA) (*Li et al., 2015*), and the rainy season as the main rainfall period is the peak flood period of the wetland (please see the 'Overview of the study area' in the article). Therefore, this article assumes that the main cause of the wetland flood, which means that the OWA of the wetland increases excessively, is caused by excessive AP in this region in a period of time. Establishing a hydrological statistical model based on the variation of AP and the OWA in the rainy season is helpful to reveal the hydrological regulation of Napahai karst wetland.

After comparing the prediction accuracy of different machine learning methods, this article selects the decision tree method. At the same time, considering the shortcomings of poor interpretability of machine learning methods, combined with the acquisition characteristics of hydrological data in the Napahai wetlands, we provide another method: the piecewise linear regression model. This article attempts to use precipitation and remote sensing data to predict Napahai wetlands floods by using a decision tree and a piecewise linear regression model, and to provide theoretical support for local flood management in terms of prediction accuracy and interpretability.

## MATERIALS & METHODS

### Overview of the study area

The Napahai watershed is a semi-closed watershed in the Hengduan Mountains on the southern margin of the Qinghai-Tibet Plateau (Fig. 1). The wetland is located at the end of the watershed, covering an area of nearly 30 km$^2$. Water comes from the Nachi river, Dalangchi river, Naizi river, dozens of mountain streams and outcropping springs, and the

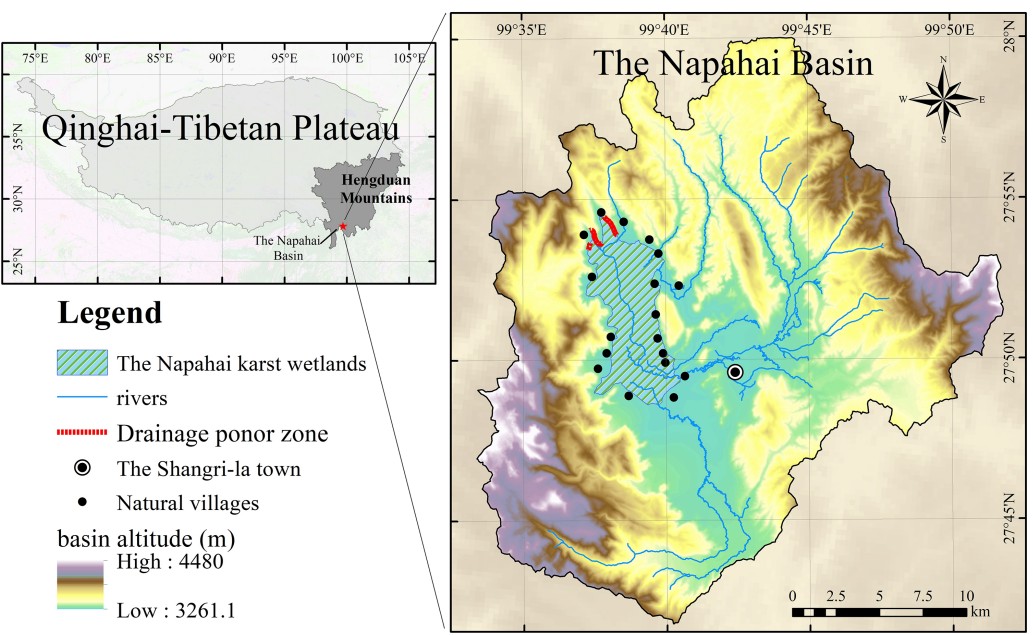

**Figure 1** The location of the Napahai karst wetland and its basin.

groundwater flows into the wetlands (*Li et al., 2013*). They are connected by a small water network together and form a complex wetland structure. Then, the water flows out to the tributaries of the Jinsha river, through the karst seepage layer distributed in the northern part of the wetlands.

Affected by the monsoon climate, the distribution of precipitation in the Napahai Basin is extremely uneven during the year. The precipitation from May to October accounts for about 85% of the annual precipitation, while the precipitation from November to the following April accounts for only 15% of the annual precipitation. The large difference in annual precipitation causes significant seasonal fluctuations in the open water surface of the wetland. The concentrated rainfall in summer is the main reason for the rapid expansion of the open water surface of the Napahai wetlands (*Li et al., 2015*). According to the statistics of the local management department, when the OWA of the wetland exceeds 1844.76 ha, the farmland will be submerged and the surrounding area will be affected. This OWA can be defined as the area threshold for flood disasters in the Napahai wetlands.

According to previous research on the Napahai wetlands, since 2000, the OWA has exceeded 1844.76 ha four times. The OWA of the Napahai Wetland in the rainy season of 2018 reached 2573.57 ha, which caused disasters to 518 surrounding residents, submerged farmland for 162 ha, causing great property losses (*Napahai Nature Reserve Administration, 2020*).

At present, due to the lack of long-term regular hydrological monitoring, such as the main river flow and water level data, only the measured data such as precipitation and the OWA data interpreted from remote sensing images can be used in the flood prediction and simulation of the Napahai Wetlands.

## Research approach

This article aims to predict the OWA by daily precipitation data of the Napahai wetland during the rainy season (May–October) from 1987 to 2018 so as to achieve the purpose of flood prediction. The reason why only the rainy season data is considered here is because the rainy season in the summer is the main source of flooding in the Napahai wetland. An OWA is predicted instead of a wetland water level due to the fact that the conventional water level data cannot be obtained, but remote sensing image data can provide OWA data.

A total of 47 OWA data of the Napahai wetland during the rainy season can be used from 1987–2018, which has the characteristic of non-equidistant time series. The routine modelling methods such as ARMA (autoregressive moving average model), ARIMA (autoregressive integrated moving average model), *etc.*, for the usual equidistant time series data cannot be applied here, but we could still borrow the idea of difference to deal with time series data. The reason why time series modelling needs to be done by difference operation is to avoid the pseudo-correlation caused by non-stationary time series. The difference in two adjacent OWA is related to the AP between the two observations and might also be related to the time interval (TI) of the two observations. Thus, we obtained the dependent and independent variables and Fig. 2 provides the data flow acquisition for the variables.

In this article, AP and TI are used to predict the difference of OWA, and the piecewise linear regression and decision tree methods are selected. The reason for choosing the piecewise linear model is based on the descriptive analysis followed that discovers that the OWA difference has a piecewise linear relationship with TI. The decision tree is chosen because it is the most accurate method to predict this data set among the most common machine learning methods such as random forest, neural network *etc.* after comparing them.

## Original data and preprocessing

(1) Extracting the Open Water Surface (OWS) of the Napahai Wetland through Landsat imageries

To ensure the consistency of data spatial accuracy, the remote sensing data used in this study are all Landsat TM (thematic mapper)/ETM+ (enhanced thematic mapper plus)/OLI (operational land imager) data. The data sources are: China Remote Sensing Data Network (http://eds.ceode.ac.cn) and USGS (United States Geological Survey) Image Database (http://glovis.usgs.gov).

A total of 47 Landsat images covering the Napahai wetlands with cloud cover less than 5% in the rainy season from 1987 to 2018 were collected. Among them, 30 scenes are Landsat4/5 TM, 6 scenes are Landsat7 ETM+ SLC (scan lines corrector)-on, three scenes are Landsat7 ETM+ SLC-off, and eight scenes are Landsat8 OLI (as shown in Table 1). We performed the geometric correction (the datum is WGS-84, UTM projection zone 47N) and radiometric correction for all image data. The modified normalized difference water index (MNDWI) method was used to interpret and extract the OWS of the wetlands to obtain its OWA in different periods. Some studies have shown that the MNDWI method can stably and effectively identify the open water landscape (*Xu, 2006*). The calculation

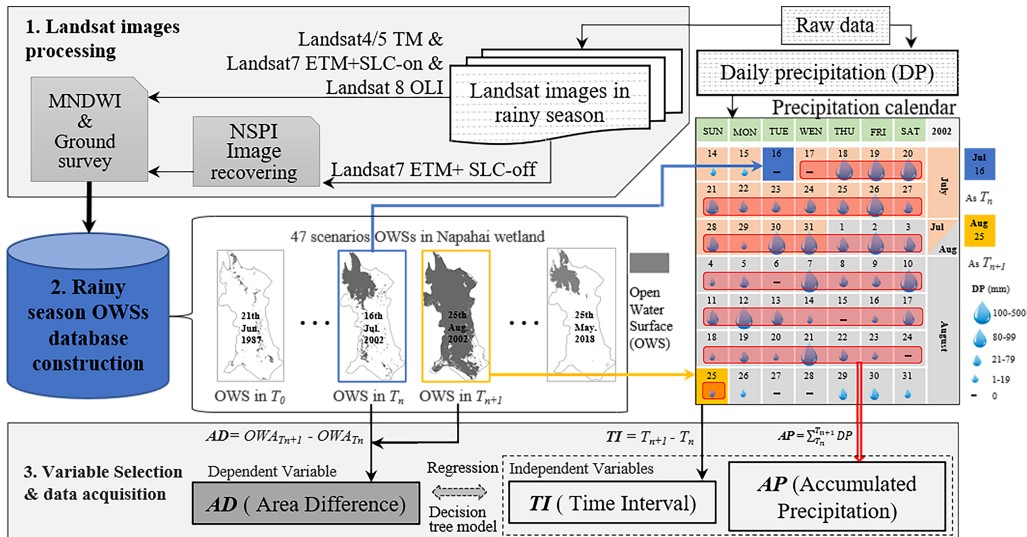

**Figure 2** **The flowchart of how the variables acquired from the Landsat images and daily precipitation data.** The first independent variable, *TI* represented the difference in days between the acquisition time of the *Nth* image and the following image. For example, The *TI* between Aug 25 and Jul 16 in the precipitation calendar was 40 days. The second independent variable, *AP* represented the accumulated DP between $T_n$ and $T_{n+1}$. The dependent variable, *AD* was calculated by the OWA different between the OWS in $T_n$ and $T_{n+1}$. OWA is the area of OWS.

method is shown in Eq. (1):

$$MNDWI = \frac{Green - MIR}{Green + MIR}. \tag{1}$$

In Eq. (1), *Green* represents the green light band in remote sensing data; *MIR* represents the mid-infrared band in remote sensing data.

Furthermore, after May 31, 2003, the SLC on the Landsat 7 ETM+ airborne scan line corrector failed, and the subsequent images (*i.e.,* ETM + (SLC -off)) lost data stripes. In this study, all Landsat 7 ETM + (SLC -off) image data were repaired by the NSPI method (*Chen et al., 2011*; *Li et al., 2015*).

In order to improve the accuracy of remote sensing interpretation, this study carried out ground surveys at the same time according to the shooting time of remote sensing images. We used an RTK (Real Time kinematic) GNSS (Global Navigation Satellite System) to collect ground landscape information. A total of 4617 points were collected within the area (as shown in Fig. 3). The sampling point density is 146.6 points/km². Within the collection of the landscape information and the surface features extracted by the MNDWI index, we determined the threshold for interpreting OWS and non-OWS as −0.31. The confusion matrix method was used to test the interpretation accuracy of the extracted OWS, the accuracy rate was 88.2%, the precision rate was 90.1%, and the Kappa coefficient was 0.76, indicating that the interpretation results were highly consistent with the real situation.

(2) The OWA of Napahai wetlands

**Table 1  Landsat-derived area of OWA in Napahai Wetlands in the rainy season.**

| ID | Time phase | Area (ha) | ID | Time phase | Area (ha) |
|----|------------|-----------|----|------------|-----------|
| 1  | 1987-06-21 | 156.42    | 25 | 2005-05-29 | 66.03     |
| 2  | 1990-05-12 | 152.09    | 26 | 2005-09-10 | 1094.54   |
| 3  | 1990-06-13 | 321.24    | 27 | 2006-05-16 | 178.76    |
| 4  | 1992-06-02 | 232.97    | 28 | 2006-07-27 | 403.77    |
| 5  | 1992-08-05 | 721.11    | 29 | 2007-10-26 | 1676.63   |
| 6  | 1995-05-10 | 210.14    | 30 | 2008-07-08 | 214.83    |
| 7  | 1996-10-19 | 310.99    | 31 | 2008-10-12 | 1490.8    |
| 8  | 1997-05-15 | 62.18     | 32 | 2009-05-08 | 207.66    |
| 9  | 1997-09-20 | 1107.86   | 33 | 2009-07-19 | 480.69    |
| 10 | 1998-09-07 | 1639.57   | 34 | 2010-06-12 | 235.74    |
| 11 | 1999-08-01 | 683.8     | 35 | 2010-07-30 | 1598.12   |
| 12 | 2000-09-20 | 1729.45   | 36 | 2010-08-07 | 1968.66   |
| 13 | 2000-10-06 | 972.18    | 37 | 2011-05-06 | 912.15    |
| 14 | 2000-10-14 | 652.09    | 38 | 2011-08-10 | 715.88    |
| 15 | 2000-10-22 | 476.94    | 39 | 2011-09-27 | 462.87    |
| 16 | 2002-05-05 | 197.05    | 40 | 2013-05-27 | 215.46    |
| 17 | 2002-07-16 | 780.24    | 41 | 2013-08-15 | 323.28    |
| 18 | 2002-08-25 | 2680.3    | 42 | 2015-05-01 | 156.78    |
| 19 | 2002-09-20 | 2890.14   | 43 | 2016-05-03 | 477.18    |
| 20 | 2002-10-02 | 2221.29   | 44 | 2016-08-07 | 1739.25   |
| 21 | 2002-10-28 | 1809.9    | 45 | 2017-05-22 | 505.89    |
| 22 | 2003-06-01 | 122.9     | 46 | 2017-06-07 | 712.26    |
| 23 | 2004-07-05 | 330.71    | 47 | 2018-05-25 | 414.81    |
| 24 | 2004-09-15 | 1702.58   |    | –          |           |

The 47 scenarios of the Napahai Wetland OWA in the rainy season from 1987 to 2018 interpreted according to Landsat images are shown in Table 1, where the sensor represents the different sensors carried by the series of Landsat satellites. In Fig. 4, the OWA in the rainy season from 1987 to 2018 fluctuated greatly, and the largest OWA appeared in 2002.

(3) Daily precipitation data

Napahai Wetland is close to Shangri-la, therefore we used the daily precipitation monitoring data from the Shangri-la meteorological station as the main data source.

The time series of annual precipitation and the average monthly precipitation of the wetland is shown in Fig. 5. The annual maximum rainfall in the Napahai wetland occurred in 2002 with 854.1 mm, followed by 817.6 mm in 2000. The rainfall was intense in July and August and steady from November to April.

## Empirical model construction

(1) Data transformation

After organizing the original data, we generated two variables that had a strong impact on area difference (AD): AP and TI. The description of each variable is shown in Table 2, and the data for each variable are shown in Table 3. TI depends on the acquisition time of
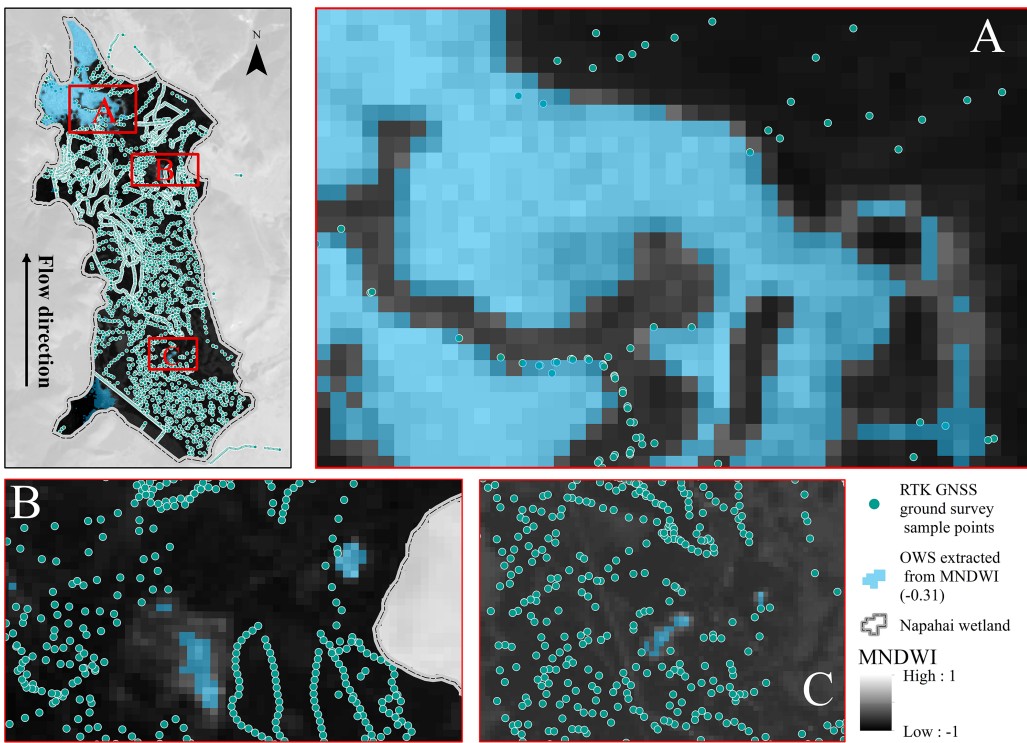

**Figure 3** Extracted open water surface (OWS) by MNDWI based on the Landsat imageries and the real-time ground survey to determine the OWS interpretation threshold value of −0.31. The MNDWI is a value between −1 and 1. With the 4,617 points RTK GNSS ground survey of OWS and non-water, when a pixel value was greater than −0.31 in the research area, it would be identified as OWS.

**Table 2** Variable description of the difference data.

| Variable type | Variable name (unit) | Value range | Description |
|---|---|---|---|
| Dependent variable | AD (ha) | [ −1687.00, 1900.05] | According to the order of the observation time, the area difference of open water between two adjacent observations. |
| Independent variables | AP (mm) | [0, 1869.8] | According to the order of the observation time, the sum of the daily precipitation of two adjacent observations. |
| | TI (day) | [8, 1056] | According to the order of the observation time, the time interval between two adjacent observations. |

**Notes.**

Abbreviations: AD, area difference; AP, accumulated precipitation; TI, time interval.

each remote sensing data point, and because the visible remote sensing data are strongly affected by cloud cover, the TI among some samples was relatively large.

(2) Descriptive analysis

In order to determine the effect of AP and TI on AD, we constructed the corresponding scatter plots, which are shown in Fig. 6. It was obvious that there was an approximately linear positive correlation between AP and AD, while the relationship between TI and

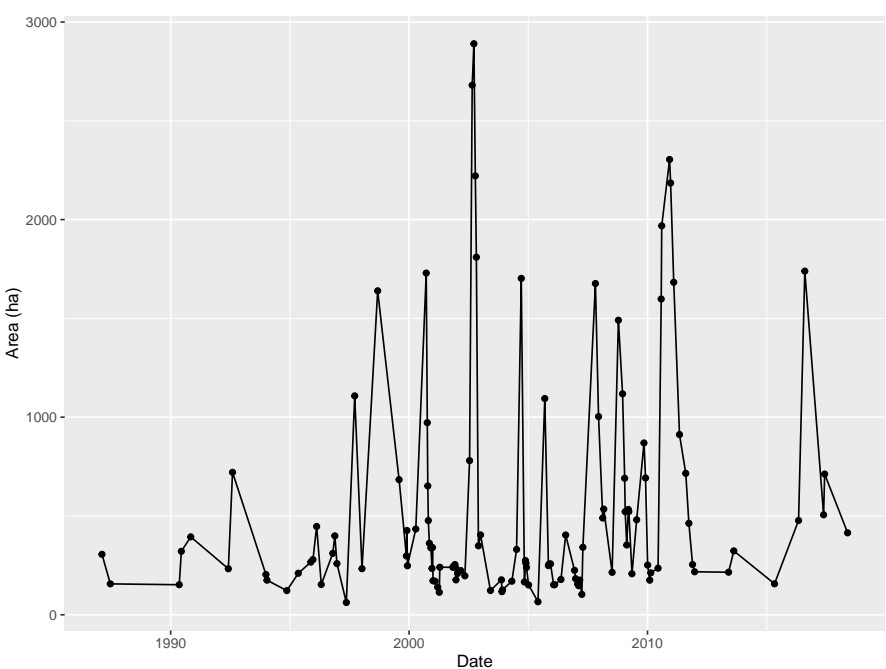

**Figure 4  Time series graph for open water area (OWA) from 1987 to 2018.**

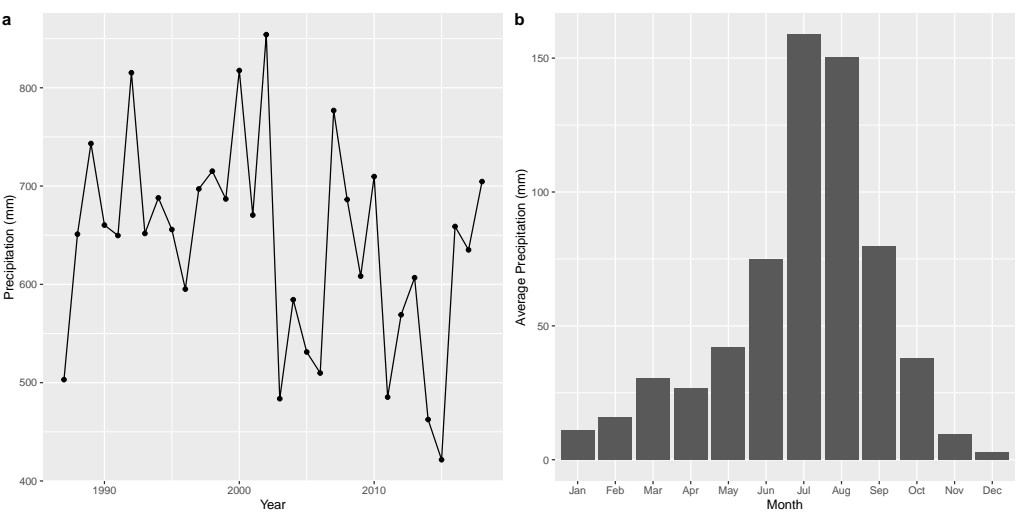

**Figure 5  Characteristic description for precipitation from 1987 to 2018.**

AD was somewhat complex, *i.e.,* a piecewise linear correlation, and the inflection point occurred at approximately 182 d.

(3) Model selection

In theory, we can choose all kinds of models for model candidates here, such as the linear regression model, non-linear regression model and all kinds of machine learning methods.

**Table 3  The corresponding data of Table 2.**

| Time phase | AD (ha) | AP (mm) | TI (day) | Time phase | AD (ha) | AP (mm) | TI (day) |
|---|---|---|---|---|---|---|---|
| 1987-06-21 | | – | | 2005-05-29 | −1636.55 | 172.4 | 256 |
| 1990-05-12 | −4.33 | 1869.8 | 1056 | 2005-09-10 | 1028.51 | 354.5 | 104 |
| 1990-06-13 | 169.14 | 98.4 | 32 | 2006-05-16 | −915.78 | 137.6 | 248 |
| 1992-06-02 | −88.27 | 1359.8 | 720 | 2006-07-27 | 225.01 | 251.4 | 72 |
| 1992-08-05 | 488.15 | 264.8 | 64 | 2007-10-26 | 1272.86 | 921.7 | 456 |
| 1995-05-10 | −510.97 | 1792.8 | 1008 | 2008-07-08 | −1461.80 | 310.4 | 256 |
| 1996-10-19 | 100.85 | 1170.5 | 528 | 2008-10-12 | 1275.97 | 349.9 | 96 |
| 1997-05-15 | −248.81 | 86.5 | 208 | 2009-05-08 | −1283.14 | 148.7 | 208 |
| 1997-09-20 | 1045.68 | 548.2 | 128 | 2009-07-19 | 273.03 | 208 | 72 |
| 1998-09-07 | 531.71 | 702.5 | 352 | 2010-06-12 | −244.95 | 460.6 | 328 |
| 1999-08-01 | −955.77 | 433.8 | 328 | 2010-07-30 | 1362.39 | 248.4 | 48 |
| 2000-09-20 | 1045.66 | 1098.6 | 416 | 2010-08-07 | 370.54 | 52.7 | 8 |
| 2000-10-06 | −757.28 | 29 | 16 | 2011-05-06 | −1056.51 | 329.4 | 272 |
| 2000-10-14 | −320.08 | 0 | 8 | 2011-08-10 | −196.27 | 268.6 | 96 |
| 2000-10-22 | −175.16 | 0 | 8 | 2011-09-27 | −253.01 | 117.2 | 48 |
| 2002-05-05 | −279.89 | 779.2 | 560 | 2013-05-27 | −247.41 | 692.5 | 608 |
| 2002-07-16 | 583.19 | 319.9 | 72 | 2013-08-15 | 107.82 | 293.4 | 80 |
| 2002-08-25 | 1900.05 | 335.4 | 40 | 2015-05-01 | −166.50 | 748.1 | 624 |
| 2002-09-20 | 209.85 | 73 | 26 | 2016-05-03 | 320.40 | 452.4 | 368 |
| 2002-10-02 | −668.85 | 50 | 22 | 2016-08-07 | 1262.07 | 378.2 | 96 |
| 2002-10-28 | −411.39 | 8.9 | 16 | 2017-05-22 | −1233.36 | 266.7 | 288 |
| 2003-06-01 | −1687.00 | 93 | 216 | 2017-06-07 | 206.37 | 44.3 | 16 |
| 2004-07-05 | 207.81 | 626.8 | 400 | 2018-05-25 | −297.45 | 584.2 | 352 |
| 2004-09-15 | 1371.87 | 918.5 | 72 | | | – | |

Based on descriptive analysis above, it can be observed that the linear regression model is inappropriate here, and the piecewise linear regression might be a reasonable model. For the machine learning methods, we tried the decision tree, bagging, random forests and support vector machines. In the end, we found that the simple method "decision tree" has the best prediction accuracy for this dataset. Regarding the comparison between the piecewise linear regression model and decision tree, they have different advantages and disadvantages. The details can be found in the section on the model comparison. In this article, we show the result details of the piecewise linear model and decision tree.

(4) Prediction accuracy

In order to evaluate a model's prediction accuracy, we consider the following evaluation indicators.

Goodness of fit: $R^2$

$$R^2 = \frac{\sum_{i=1}^{n}(\hat{y}_i - \overline{y})^2}{\sum_{i=1}^{n}(y_i - \overline{y})^2}. \tag{2}$$
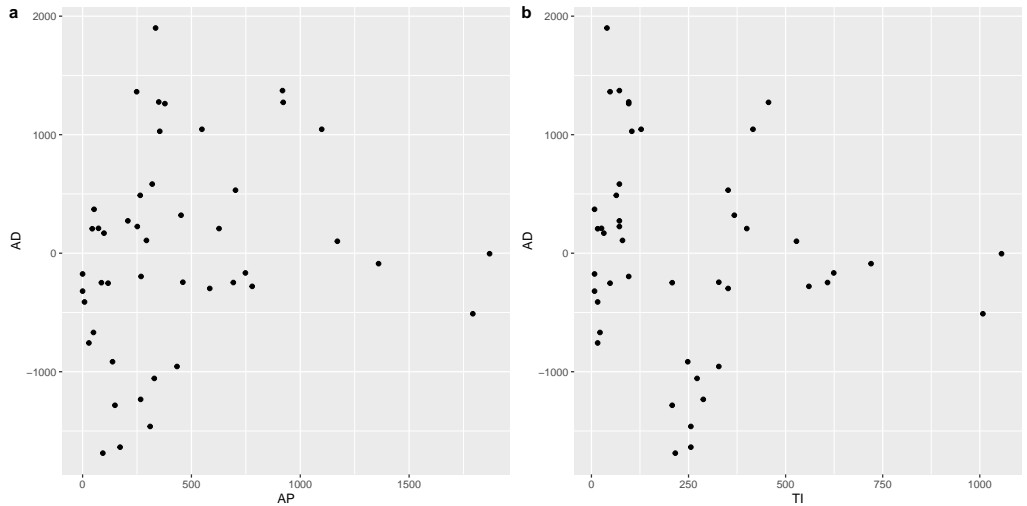

**Figure 6** Scatter plot between AP and AD, TI and AD respectively.

Adjusted $R^2$:

$$R_A^2 = 1 - (1 - R^2) \frac{n-1}{n-p-1}. \tag{3}$$

Mean Square Error:

$$MSE = \frac{1}{n} \sum_{i=1}^{n} (\hat{y}_i - y_i)^2. \tag{4}$$

Note $n, p, y_i, \hat{y}_i$ represent the sample size, number of covariates, *i-th* observation of the dependent variable and its fitting value respectively. $\bar{y}$ is the average value of all of the observations of the dependent variable. $R^2$ is used to evaluate the fitting effect of the regression model; it ranges from 0 to 1, and a higher value indicates a better model fit. However, there is a tendency to encourage more explained variables even meaningless explained variables. The adjusted $R^2$ makes up for the shortcoming (*Rencher & Schaalje, 2008*). The MSE will be small if the predicted responses are very close to the true responses and will be large if for some of the observations, the predicted and true responses differ substantially (*James et al., 2013*).

## RESULTS

### Model results

It was observed that the relationship between AD and TI was segmented (see Fig. 6B), with 182 d (half a year) as the inflection point, and two linear regression models were established afterward, corresponding to the fitting of ≤182 d and >182 d. The results are shown in Table 4.

Table 4 shows that for every one mm increase in AP, AD increased by 2.41 ha, whether TI was longer or shorter than 182 d. This indicated that the impact of AP on AD was

**Table 4  Results for the linear regression model.**

| | TI ≤ 182 | | | | TI > 182 | | |
|---|---|---|---|---|---|---|---|
| Variable | Regression coefficients | *P*-value | Remark | Variable | Regression coefficients | *P*-value | Remark |
| Intercept | −150.73 | 0.36 | | Intercept | −382.56 | 0.25 | |
| AP | 2.41 | <0.001 | | AP | 2.41 | 0.004 | |
| | | | | TI | −3.66 | 0.03 | |
| *F* test | | <0.001 | R² = 0.51 | *F* test | | <0.001 | R² = 0.43 |

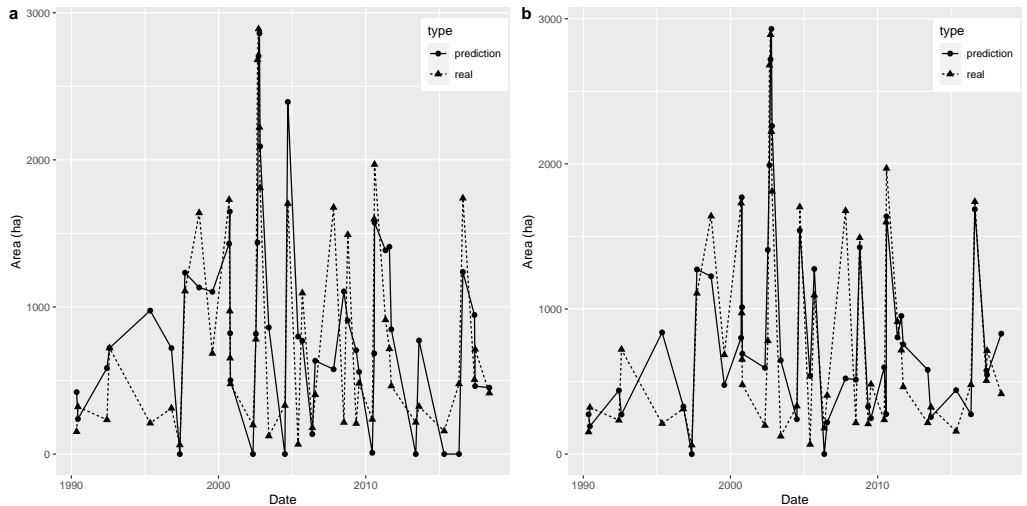

**Figure 7  (A–B) Prediction graph for the piecewise linear regression model and the decision tree.**

independent of TI. In addition, when TI ≤ 182 d, TI had no significant effect on AD, and therefore it did not appear in the final model. When TI > 182 d, TI had a significant negative correlation with AD, shown in Table 4; for every 1 d increase in TI, AD decreased by 3.66 ha.

Therefore, we predicted the AD value through the above model, and then used the nearest OWA extracted by remote sensing images to predict the OWA on a specific observation date. Based on a comparison of the predicted OWA and the extracted OWA, which are presented in Fig. 7A, we found a certain difference between the two series in general. However, the model produced a relatively accurate prediction for 2002, which was when the OWA reached the historical peak, therefore this model has a certain reference value for flood forecasting.

The results of the decision tree are shown in Fig. 8. In total, there were 46 samples, and the average AD was 5.62 ha. The whole data set was divided into two parts with a TI of 168 days as the dividing point. There were 23 samples whose TI was shorter than 168 days, and the corresponding average value of AD was 396 ha. When the AP was less than 307 mm, there were 16 samples, and the AD was predicted to be 39.4 ha. In contrast, when the AP exceeded 307 mm, the AD was predicted to be 1210 ha. There were 23 samples whose TI

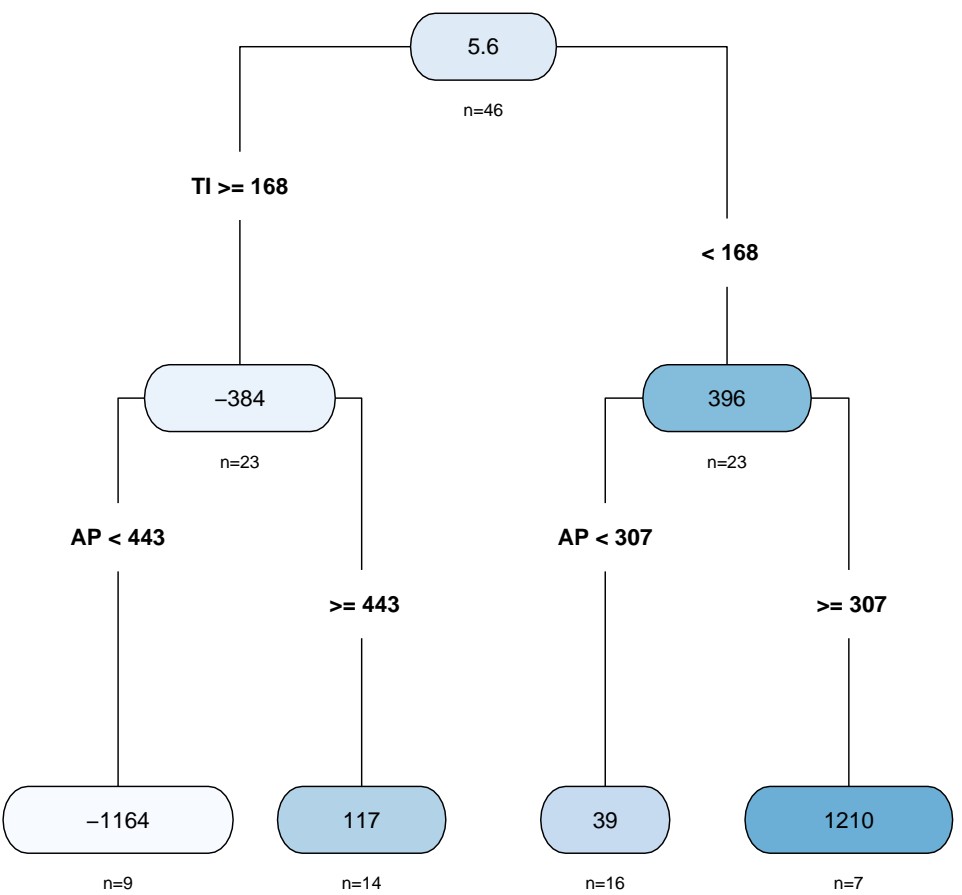

**Figure 8  The decision tree plot.**

exceeded 168 days; the corresponding AD was −384 ha; among them, only nine samples had AP values less than 443 mm, and the corresponding AD was predicted to be −1164 ha; otherwise, it was predicted to be 117 ha.

The comparison between the OWA predicted by the decision tree model and the extracted OWA is shown in Fig. 7B (note: when the predicted value was negative, we set it to 0). Compared with the piecewise linear regression model, the accuracy of this method was relatively high. In addition, there was little difference between the predicted OWA and the extracted OWA for the largest flood in 2002, which means the decision tree model was accurate in terms of flood simulation.

## Model comparison

(1) Comparison of prediction accuracy

The $R^2$, adjusted $R^2$ and MSE results are shown in Table 5. Both the piecewise linear regression model and the decision tree have a moderate level of fitting effect. For example, the decision tree model could explain approximately 69% variation of the sample AD for this dataset. Both methods have relatively large MSE. There might be two reasons for that.

| | Piecewise linear regression | Decision tree |
|---|---|---|
| **Table 5** **Comparison of model fitting effect.** | | |
| $R^2$ | 0.58 | 0.69 |
| Adjusted $R^2$ | 0.56 | 0.68 |
| MSE | 411325.04 | 492566.04 |

First, both methods have difficulty in predicting all the observations accurately which yield large MSE. Second, the sample size is small, which is an added challenge for prediction accuracy. Obviously, the decision tree model has better $R^2$, adjusted $R^2$ but worse MSE. We might understand the results in the way that the decision tree has less bias for this dataset, but the piecewise linear model is more robust here.

(2) Comparison of the interpretability of the models

Regarding the fitting results of the linear model, each regression coefficient has a definite meaning in practice. In particular, the piecewise linear regression coefficients in Table 4 were explained before (see 'Model results' section) and are helpful for local managers in Napahai to understand how AP and TI influence AD. On the contrary, the decision tree is a type of machine learning method; its modelling is mainly based on an algorithm, and it is almost impossible to interpret its results.

In conclusion, the piecewise linear model has a relative lower prediction accuracy, but a better interpretability and robustness compared with the decision tree. Considering that they both have advantages and disadvantages, we chose to keep both methods as our final models.

## DISCUSSION

### Decision analysis

In order to help the water managers to predict the flood in the Napahai wetland, the article provides two schemes. The decision tree is better in terms of prediction accuracy, and the piecewise linear regression is better in terms of comprehensibility and robustness.

In fact, we can combine the two models' results to give a simpler decision scheme. First, according to the decision tree, we verified whether AP is greater than 307 mm to make a preliminary judgment on the flood risk. If not, it is judged as low risk, otherwise it is high risk, and the manager needs to predict whether OWA will exceed 1844.76ha further based on the piecewise linear regression model.

### Limitations

In this article, the prediction analysis method has some limitations, mainly related to the following two aspects: (a) Prediction accuracy is not very high. The fundamental reason is the difficulty of data acquisition. Data access poses a challenge concerning other factors that may affect flood and equidistant OWA time series data. Data acquisition is one of challenges for Karst flood predictions (*Hartmann et al., 2014*). (b) This article only simulates the OWA of the Napahai wetland, and it will be better if it can be extended to the

two-dimensional open water distribution (*Wen et al., 2013*; *Seleem et al., 2022*). We will consider this in the future when we can obtain more data.

## Comparison with similar studies

Precipitation is a core element of the terrestrial hydrological cycle, and therefore almost all studies related to hydrological simulation will use rainfall as the main variable of their models (*Ghasemizadeh et al., 2012*; *Kovács & Sauter, 2007*; *Basu, Morrissey & Gill, 2022*). From this point of view, this study is consistent with similar studies. Data-driven models are also widely used in research on flood prediction (*Guzman, Paz & Tagert, 2017*; *Hadid, Duviella & Lecoeuche, 2020*; *Sawaf et al., 2021*). Some studies have expressed the advantages of data-driven model compared with the physical model: it has fewer requirements for hydrological variables (*Ji et al., 2012*; *Seleem et al., 2022*), which reflects in the model building that usually has data-lacking problems. Researchers tend to use precipitation, water level, discharge and other variables for modeling with sufficient data. However, when these data are insufficient, researchers are more inclined to use multi-source remote sensing data to obtain hydrological data such as OWS and so on (*Windolf et al., 2011*; *Spence et al., 2013*; *Yang, Cai & Wang, 2018*), which is similar to the method used in this study.

Moderate-resolution imaging spectroradiometer (MODIS) remote sensing data (with a high satellite replay period: 1d and relatively low spatial resolution: 250 m) are often used to extract the OWS of the ungauged wetland. The advantage of using MODIS as the data source is that it can obtain enough OWA data within a relatively long study period, which provided sufficient samples (*Gumbricht et al., 2004*). The disadvantage is that, for relatively small wetlands, its low spatial resolution cannot accurately represent the pattern of the OWS.

In the studies of flood prediction modeling, machine learning methods were widely used. These studies gained fine prediction accuracy for the large amount of training samples (*Qian, Mohamed & Claudel, 2019*; *Basu, Morrissey & Gill, 2022*). However, these studies still cannot quantitatively explain the impact of precipitation on flooding. Therefore, in addition to the machine learning model, the traditional regression model can make up for the disadvantages of the machine learning model, which is also the reason why this article maintains both models.

## CONCLUSIONS

This article described a piecewise linear regression model and a decision tree model to provide solutions for flood predictions of the Napahai wetland based on limited and non-equidistant time series data. We combined both methods to provide an easy effective risk management for flood prediction that makes a preliminary judgment of flood risk level based on whether AP is larger than 307 mm (decision tree result), and then check if OWA is larger than the flood threshold based on piecewise linear regression model if necessary.

## FUTURE WORK

(1) Obtaining multi-source remote sensing data (such as in recent years, constantly enrich the high-resolution optical remote sensing data, as well as microwave and radar remote

sensing data; among them, the latter can effectively reduce the influence of the monsoon cloud cover for remote sensing data acquisition), extract more OWA data of Napahai wetland, and increase the number of samples for the model, so as to improve the model accuracy. (2) If sufficient OWS data of the wetland are obtained, the spatial simulation of the flood could be modeled in Napahai karst wetland along with the local meteorological data, spatial feature statistics of the OWS and the terrain model of the wetland.

**Index of Abbreviations and Notations**

**Abbreviations**

| | |
|---|---|
| **OWA** | open water area |
| **AD** | area difference |
| **AP** | accumulated precipitation |
| **TI** | time interval |
| **ARMA** | autoregressive moving average model |
| **ARIMA** | autoregressive integrated moving average model |
| **OWS** | open water surface |
| **TM** | thematic mapper |
| **ETM+** | enhanced thematic mapper plus |
| **OLI** | operational land imager |
| **USGS** | United States geological survey |
| **MNDWI** | modified normalized difference water index |
| **RTK** | real time kinematic |
| **GNSS** | global navigation satellite system |

**Notations**

| | |
|---|---|
| **d** | day |
| **ha** | hectare |

## Funding

This work was supported by the National Natural Science Foundation of China (No. 41601060). The funders had no role in study design, data collection and analysis, decision to publish, or preparation of the manuscript.

## Grant Disclosures

The following grant information was disclosed by the authors:
National Natural Science Foundation of China: 41601060.

## Competing Interests

The authors declare there are no competing interests.

## Author Contributions

- Xiao Li conceived and designed the experiments, performed the experiments, analyzed the data, prepared figures and/or tables, authored or reviewed drafts of the article, and approved the final draft.
- Jie Li conceived and designed the experiments, performed the experiments, analyzed the data, prepared figures and/or tables, authored or reviewed drafts of the article, and approved the final draft.

## Data Availability

The Open Water area data is packaged in a geodatabase (that can be opened by using ArcGIS software), and Daily precipitation data, are available in the Supplemental Files.

## Supplemental Information

Supplemental information for this article can be found online at http://dx.doi.org/10.7717/peerj.14940#supplemental-information.

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
