# Peer review of "Data-driven models for flood prediction in an ungauged karst wetland: Napahai wetland, Yunnan, China"

_PeerJ, doi:10.7717/peerj.14940_

## Round 0.1 · original submission · Major Revisions

Please consider the reviewers' comments carefully.

·

Basic reporting

The wetland flood prediction model is a topic that is increasingly interested in being studied. As the authors have explained a large part of the population lives in wetlands in this view every article in this field is an added value.

The authors have done a good job, but not enough in order to have reliable and referable results in the future. The number of initial data in this study/studies is relatively large, that is why it is necessary to explain how much of this data has been used, and at the same time which one of the initial data has the highest reliability and which ones need further interpretation.

Research ideas, Original data collection, and Empirical model construction are the essential/main part of this article, if the mentioned above parts of this paper are not explained in detail and compared with at least some similar recent work then the results may be questionable.

R square value of linear regression is 0.58 and that of decision tree 0.69, in both cases the accuracy/goodness is not high. The authors have tried to explain the reasons resulting from these findings however they are not detailed and there is not a single comparison with previous works.

The language and the text of the paper have to be improved/reorganized if there would be a decision for this article to be published.

Experimental design

Research ideas, Original data collection, and Empirical model construction are the essential/main part of this article. It is explained in the general comments the works performed in this study have been vast, especially for the last 2 decades. So the authors should have explained and taken into consideration the following.

1. Research ideas
a. What is their research idea closed to
b. Some of the articles that have used the same/similar idea
c. Some of the articles that have used the different/ideas
d. Reasoning why the authors have believed their idea could result in better findings

2. Original data collection
a. Access to original data
b. The decision to group the original data in reliable and not reliable ones
c. Methodology of processing the data before making them part of the empirical models

3. Empirical model construction
a. Selection of the empirical model (what are the fundamentals and literature review done by the authors in order to choose the 2 proposed models
b. Limitation of these models regarding the specific topic
c. What have the authors done in order to minimize the errors in the findings

Validity of the findings

It is essential to reorganize the research idea together with the empirical models and at the final stage of the research to propose a wetland flood prediction model that better fits the analyzed area rather than a comparison of the two models their selves.

Additional comments

The language and the organization of the text have to be improved/reorganized if there would be a decision for this article to be published.

Reviewer 2 ·

Basic reporting

“A karst wetland flood prediction model: A case study of the Napahai Wetland, Yunnan, China”
This study wants to simulate the open water area (OWA) of the Napahai karst wetland through remote sensing as the starting point to describe its flood characteristics using the local daily precipitation and the 1987–2018 rainy season, which maybe is interesting. However, it is very difficult to understand the words in this manuscript, so I suggest this manuscript need be polished by a fluent English speaker. Some comments as follows:
1. I think authors need rewrite the introduction. You did not collect what different researcher do, but you need conclude their research and upgrade the mechanism on karst wetland. Moreover, I even did not know what authors want to do and what your hypothesis is.
2. Method section, if authors have sampled in the Napahai Wetland? It looks you did not test and calibrate the simulated results by model.
3. It lacks of enough discussion linked with the area and water source change in in the Napahai Wetland.
4. There were very low-quality figures and tables.

Experimental design

No data descriptiton and uncentainity.

Validity of the findings

No provide

Additional comments

No

Reviewer 3 ·

Basic reporting

I think the knowledge gap needs to be better explained. Also, the authors should explain why wetland karst matters. Instead, authors more focused on wetland in general that makes the manuscript does not have strong background for the research. Limited cited literature (such as in L53) related the topics addressed confirms this flaw.
Authors also shall focus how to deliver the message clearly as in current form seem lack coherency among paragraphs.
I don’t understand what the meaning this phrase ’precipitation is a stable and reliable data source’ (L78). In all hydrological modeling, precipitation is the main input data as clearly stated in L80-81.
The last paragraph of the intro is confusing; what kind of message is like to deliver? Also, it didn’t clear where is the authors idea and which ones is not (refers to citation)
I think in the current form, the manuscript needs substantial revision on writing style and structuring the texts. Further, the manuscript lack of discussion why the research matters to give the readers a sense of worth findings, and how the findings differ with others.


L119 ‘… the once- …’ what is it?
L 125 Research ideas?
L157 Data editing or data manipulation?
L204-218 is for Method section
L221 there was not any method related to suffering from wetland flood.
L262 challenging?

Experimental design

no comment

Validity of the findings

no comments

---

## Round 0.2 · Major Revisions

The work is interesting, has a clear degree of originality, and is appropriate for publication in the journal after performing a major and very careful revision. Nevertheless, it needs some further improvements. In general, there are still some occasional grammar errors throughout the manuscript, especially the article "the," "a," and "an" are missing in many places; please make spellchecking in addition to these minor issues. The reviewer has listed some specific comments that might help the authors further enhance the manuscript's quality.

1. Specific Comments

• Overall, the Abstract section is not giving any information about methodology, results, conclusion, and recommendations as it should be. I suggest the authors to remove generic lines and present the strong statements and novelty of article. The abstract is written in qualitative sentences. It is necessary to modify and rewrite based on the most important quantity results from this research. The abstract should be redesigned. You should avoid using acronyms in the abstract and insert the work's main conclusion.

• You have used many abbreviations in the text. From this perspective, an Index of Notations and Abbreviations would be beneficial for a better understanding of the proposed work. Furthermore, please check carefully if all the abbreviations and notations considered in work are explained for the first time when they are used, even if these are considered trivial by the authors. The paper should be accessible to a wide audience. Furthermore, it will make sense to include also the notations in this index.

• The objectives should be more explicitly stated.

• The Introduction section must be written better. The research gap should be explained clearly along with the necessity for the conducted research work.

• What is the novelty of this work?

• It is better to improve your contributions which are not so clear to show the advantage of your work.

• The novelty of this work must be clearly addressed and discussed in Introduction section.

• The methodology limitation should be mentioned.
Many equations are presented in the paper, and most look OK. However, please check carefully whether all equations are necessary and whether the quantities involved are properly explained. Also, some equations need references.

• Results
• This section is well written.

• Discussion
• Overall, the discussion part is weak. The Discussion should summarize the manuscript's main finding(s) in the context of the broader scientific literature and address any study limitations or results that conflict with other published work.

• Conclusion
• Some future works should be added to your conclusion.

Reviewer 2 ·

Basic reporting

“Data-driven models for flood prediction in Napahai karst wetland”
In this revised manuscript, authors try to use precipitation and remote sensing data to predict
Napahai wetlands floods, which seem to be interesting, and also show us local flood management in terms of prediction accuracy and interpretability. However, though wetland mainly depend on local precipitation, it also correlated with runoff input and other source supplement. In addition, I can not more scientific problem and hypothesis in this manuscript, and only show some based monitor data or remote sensing data.
Some comments as follows:
1. The language on this manuscript needs to be repolished by a fluent English speaker. Please submit the manuscript with double space.
2. In introduction, authors should not conclude general question, but need focus on the related progress on the wetland are responds to climatic change and which the important scientific difficulty is. What scientific hypothesis you advanced?
3. In my opinion, the figures and tables in this manuscript need further be improved and supplemented more data to those figures and tables. I have no more comment about discussion and results, because I can not read more useful information and conclusion.

Experimental design

In methods and results, authors only provide rainfall as one indicator, but not consider runoff and temperature or other runoff input impact on wetland change. Especially, author need make a statistic on wetland area change at different period, and then discuss their driving mechanism.

Validity of the findings

Decisions are not made based on any subjective determination of impact, degree of advance, novelty or being of interest to only a niche audience.

Reviewer 3 ·

Basic reporting

my comments in pdf file

Experimental design

my comments in pdf file

Validity of the findings

my comments in pdf file

Additional comments

my comments in pdf file

Annotated reviews are not available for download in order to protect the identity of reviewers who chose to remain anonymous.

---

## Author Rebuttal · Round 0.2

Dear Sir or Madam,

We thank Professor Erion Periku and the other two anonymous reviewers for their very constructive comments and suggestions. Major revisions have been made based on these comments, especially we made a great change on section Introduction, Overview of the Area Study, Research Idea, Original Data, and Discussion. We added two new graphs and one new section called Decision Analysis. The following is our point-to-point response, and Reviewer's original comments are shown in **Bold** and *italic*.

Sincerely yours,

Authors

**Reviewer: Erion Periku**

*In basic reporting part:*

*1. …The number of initial data in this study/studies is relatively large, that is why it is necessary to explain how much of this data has been used, and at the same time which one of the initial data has the highest reliability and which ones need further interpretation.*

**Answer:** This paper aims to predict the open water area (OWA) by daily precipitation data of the Napahai wetland during the rainy season (May - October) from 1987 to 2018 so as to achieve the purpose of flood prediction. The reason why only the rainy season data is considered here is because the rainy season in summer is the main source of flood in the Napahai wetland. (Paragraph 1 in Research Idea). The original data includes OWA, date and daily precipitation. Daily precipitation was observed every day during the rainy season from 1987 to 2018 and reliable. Only 47 OWA data was available from Landsat images in the corresponding period. We also made some corrections the image data in order to get a more reliable data. Details were shown on the first and second paragraph of Original data.

*2. Research ideas, Original data collection, and Empirical model construction are the essential/main part of this article, if the mentioned above parts of this paper are not explained in detail and compared with at least some similar recent work then the*

*results may be questionable.*

**Answer:** In the new version of paper, research ideas, original data collection, and empirical model construction were explained in detail. As compared with at least some similar recent work, we would like to point out one fact: there are usually three categories modelling methods for flood prediction of Karst wetlands (details in Introduction). Physics-based distributed hydrodynamic models, data-driven models and semi-distributed models. Due to the lack of data, physics-based distributed hydrodynamic and semi-distributed models cannot be applied here. Most data-driven models for Karst wetlands mainly focused on machine learning methods. The decision tree (one of methods of machine learning) is chosen in this paper because it is the most accurate method to predict this data set among the common machine learning methods such as random forest, neural network etc. after comparing them. We mentioned the above information in detail in the part Introduction and Research Idea.

*3. R square value of linear regression is 0.58 and that of decision tree 0.69, in both cases the accuracy/goodness is not high. The authors have tried to explain the reasons resulting from these findings however they are not detailed and there is not a single comparison with previous works.*

**Answer:** In the new Discussion part of the paper, we explained the fundamental reason for the relatively low goodness of fit is the difficulty of data acquisition. Data access poses a challenge concerning the other factors that may affect flood and equidistant OWA time series data in this paper. Actually, data acquisition is one of the challenges for Karst flood predictions based on what's mentioned in some references. The main difference between our work and previous works is we tried to afford modelling methods for non- equidistant time series data due to data access difficulty. We think the R square is good enough in this paper based on the current data quality.

*4. The language and the text of the paper have to be improved/reorganized if there would be a decision for this article to be published.*

**Answer:** Thank you for your suggestion. We have polished our language for the new version of paper.

*In experimental design part:*

*1. Research ideas*

*a. What is their research idea closed to*

*b. Some of the articles that have used the same/similar idea*

*c. Some of the articles that have used the different/ideas*

*d. Reasoning why the authors have believed their idea could result in better findings*

**Answer:** Due to the challenge of data access, we have a total of 47 OWA data of the Napahai wetland during the period we considered, which has the characteristic of non-equidistant time series. The routine modelling methods for the usual equidistant time series data cannot be applied directly here, but we borrowed the idea of difference to deal with time series data and generated dependent variable AD and independent variables TI and AP for model construction. Based on what we know, all the references to use data-driven models for Karst flood prediction are related with equidistant time series data. We are the first one trying to construct data-driven model for non-equidistant data. The details can be found in the new version Introduction and Research Idea part of the paper.

*2. Original data collection*

*a. Access to original data*

*b. The decision to group the original data in reliable and not reliable ones*

*c. Methodology of processing the data before making them part of the empirical models*

**Answer:** The original data includes daily precipitation acquired from local meteorological station and 47 scenes of Landsat images in rainy season from 1987 to 2018. The daily precipitation data is reliable.

Most open water surface (OWS) data are highly reliable because of data source consistency, high data quality (cloud cover less than 5%, etc.), unifying preprocessing, and ground validation. The 3 scenes of OWS extracted from the Landsat7 ETM+ slc-off data brought some uncertainty to the original data caused by the scan line corrector failure of the satellite. But the images were repaired by the NSPI method, which was a widely accepted method for Landsat7 ETM+ slc-off image repairing. The figure.1 below shows an example of the image before and after the repairing of the images in

Napahai wetland. Compared with other OWS data, the reliability of these 3 scenes of OWS were relatively low.

[Figure]

Figure1. one example of image before and after the repairing

We used MNDWI method to extract the OWS. When using MNDWI to extract OWS, it is necessary to detect the boundary of the wetland open water through field survey (as shown in figure 3 of the new version of the manuscript), for determining the threshold value of OWS extraction. The OWA is the area of the OWS, and can be read directly from the OWS. More details can be found in the Original Data.

*3. Empirical model construction*

*a. Selection of the empirical model (what are the fundamentals and literature review done by the authors in order to choose the 2 proposed models*

*b. Limitation of these models regarding the specific topic*

*c. What have the authors done in order to minimize the errors in the findings*

**Answer:** In the literature review part, we introduced there were three categories modelling methods for Karst wetland. Physics-based distributed hydrodynamic models, data-driven models and semi-distributed models. Due to the lack of data, physics-based distributed hydrodynamic and semi-distributed models cannot be applied here. Considering the characteristic of non-equidistant time series of the original data, we

made the difference of the data at first and got the variable AD, TI and AP. After that, we chose two models: piecewise linear regression model and decision tree. Our logic for model selection is as followed: classic regression model and machine learning are main types of data-driven models. Usually, machine learning methods have better prediction than classic regression methods, but classic regression methods could provide easy and clear interpretation based on the regression results. Our discussion part also verified it. In practice, we care about both prediction accuracy and result interpretation. Therefore, we decided to choose one classic regression method and one machine learning method simultaneously. Based on the descriptive analysis, we could see clearly there is no linear correlation between AD and TI, but a piecewise linear correlation. Therefore, we chose piecewise linear regression model as one model in the end. As for the decision tree, we explained in the Research Idea part that it's the most accurate method to predict this data set among the common machine learning methods. In the Decision Analysis part, we proposed one decision strategy combining the two models in application.

***In validity of the findings part:***

1. ***It is essential to reorganize the research idea together with the empirical models and at the final stage of the research to propose a wetland flood prediction model that better fits the analyzed area rather than a comparison of the two models their selves.***

**Answer:** We reorganized Research Idea, Empirical Models and added one section called Decision Analysis. Besides prediction accuracy, we also care about result interpretation in practice. This is one reason why we keep the two models. Another reason is we combined the two model to propose one decision strategy for local water manager for making decisions.

***In Additional comments part:***

1. ***The language and the organization of the text have to be improved/reorganized if there would be a decision for this article to be published.***

**Answer:** We did some effort on the language and organization in the new version.

**Reviewer #2**

*In basic reporting part:*

*1. …However, it is very difficult to understand the words in this manuscript, so I suggest this manuscript need be polished by a fluent English speaker.*

**Answer:** Thank you for your suggestion. The manuscript has been polished by a native English speaker.

*2. I think authors need rewrite the introduction. You did not collect what different researcher do, but you need conclude their research and upgrade the mechanism on karst wetland. Moreover, I even did not know what authors want to do and what your hypothesis is.*

**Answer:** Thank you for your suggestion. We did rewrite the introduction especially for the literature review. We proposed our question at the end of introduction.

*3. Method section, if authors have sampled in the Napahai Wetland? It looks you did not test and calibrate the simulated results by model.*

**Answer:** In the study, we tried to build two data-driven models for the flood prediction rather than using a well-developed hydrological model. Because the karst wetland was lack of basic monitoring and hydrological data, the only adaptive data for the model building in the study area were the OWA derived from Landsat remote sensing images and the local daily precipitation. So based on these data, a piecewise linear regression model, for its better interpretability, and a decision tree model, for its better prediction, were built to supporting the decision making of the local water managers. Therefore, there is no need to calibrate the results by models.

In Results section, we did a comparison of the predicted values and the extracted values (which could be seen as the real measured value), shown in figure 9 and figure 11 (in revised version of the manuscript). These parts could be seen as the test of the models.

*4. It lacks of enough discussion linked with the area and water source change in the Napahai Wetland.*

**Answer:** We revised the section Overview of the Study Area this time and added more discussion related with the area and water source change in the Napahai Wetland.

*5. There were very low-quality figures and tables.*

**Answer:** We added two figures and made some change to the tables.

*In Experimental design part:*

1. *No data descriptiton and uncertainity.*

**Answer:** We revised the section Original Data, and explained in detail how we accessed data, which data is reliable and which not, and what kind of methods we used to improve data reliability etc. Data description can be found in Original Data, Descriptive Analysis of Empirical Model Construction.

**Reviewer #3**

*In basic reporting part:*

*1. I think the knowledge gap needs to be better explained. Also, the authors should explain why wetland karst matters. Instead, authors more focused on wetland in general that makes the manuscript does not have strong background for the research. Limited cited literature (such as in L53) related the topics addressed confirms this flaw.*

**Answer:** Based on the above questions, we rewrote our Introduction, Research Idea. Please check the new revision.

*2. Authors also shall focus how to deliver the message clearly as in current form seem lack coherency among paragraphs.*

**Answer:** Thank you for your suggestion. In the new version of the manuscript, we tried to improve coherency among paragraphs and hired an English speaker to polish our language.

*3. I don't understand what the meaning this phrase 'precipitation is a stable and reliable data source' (L78). In all hydrological modeling, precipitation is the main input data as clearly stated in L80-81.*

**Answer:** The above sentence was deleted in the new version of manuscript.

*4. The last paragraph of the intro is confusing; what kind of message is like to deliver? Also, it didn't clear where is the authors idea and which ones is not (refers to citation). I think in the current form, the manuscript needs substantial revision on writing style and structuring the texts. Further, the manuscript lack of discussion why the research matters to give the readers a sense of worth findings, and how the findings differ with others.*

**Answer:** We rewrote the Introduction, Research Idea and Original Data. Based on what we know, we are the first one to build data-driven models for non-equidistant time series data of Karst wetland flooding prediction. We tried to afford to the readers some strategy for non-equidistant time series modelling due to the data acquisition challenge which in common for Karst flood prediction.

*5. L119* '⋯ *the once-* ⋯' *what is it?*

*L 125 Research ideas?*

*L157 Data editing or data manipulation?*

*L204-218 is for Method section*

*L221 there was not any method related to suffering from wetland flood.*

*L262 challenging?*

**Answer:**

1. L119 '… the once- …' what is it? Means it happened once in twenty years. We deleted it in the new version.

2. Research Idea.

3. We changed the title to "Data transformation".

4. We moved the definition of $R^2$ to the Empirical Model Construction part in Materials & Methods section.

5. We revised the discussion section.

6. We revised the discussion section.

---

## Round 0.3 · Minor Revisions

The review of your paper is now complete, the Reviewers' reports are below. As you can see, the Reviewers present important points of criticism and a series of recommendations. We kindly ask you to consider all comments and revise the paper accordingly in order to respond fully and in detail to the Reviewers' recommendations. If this process is completed thoroughly, the paper will be acceptable.

Reviewer 3 ·

Basic reporting

In the revised version, authors already answered and responded to my concerns, and now it is more readable.

Experimental design

Ok

Validity of the findings

ok

Additional comments

minor:

L24 'Without having data on the water level, velocity, discharge and so on..' change to ‘in ungauged wetland/condition
L26 remove ‘moreover’
L46 US dollars to US$
L70-73 merge with next paragraph

---

## Round 0.4 · accepted · Accept

I congratulate the authors for the effort put into this paper! The manuscript is significantly improved; therefore, I recommend accepting it in its current form!

Reviewer 3 ·

Basic reporting

okay

Experimental design

okay

Validity of the findings

okay

Additional comments

for L24, just use wetland or condition, not both